# A Novel Evaluation Method of Construction Homogeneity for Asphalt Pavement Based on the Characteristic of Component Distribution

**DOI:** 10.3390/ma15207284

**Published:** 2022-10-18

**Authors:** Ke Zhang, Guangliang Wei, Wei Xie, Baocheng Yang, Wenlong Li, Yaofei Luo

**Affiliations:** 1College of Information Engineering, Fuyang Normal University, Fuyang 236041, China; 2School of Civil Architecture, Anhui University of Science & Technology, Huainan 232001, China; 3School of Civil Engineering and Architecture, Zhengzhou University of Aeronautics, Zhengzhou 450046, China

**Keywords:** asphalt pavement, component, construction homogeneity, evaluation method, horizontal heterogeneity coefficient, vertical heterogeneity coefficient

## Abstract

To effectively evaluate the construction homogeneity of asphalt pavement, the tomography image of a core sample of asphalt pavement was obtained via industrial computed tomography (CT) equipment. According to the characteristics of CT images, an improved separation algorithm based on annular partition and Nobuyuki Otsu (OTSU) threshold segmentation was proposed. Based on the distribution of aggregates, voids and asphalt mortar, and the area ratio of each part in the CT images inside the pavement, a novel evaluation method for the distribution homogeneity of asphalt pavement components was put forward, and the validity of the evaluation index was also verified. The results show that the aggregates, voids and asphalt mortar in CT images can be effectively segmented by annular partition combined with the OTSU threshold separation algorithm. By superimposing the segmented image on the original image, the segmentation and identification effects of aggregates, voids and asphalt mortar in the CT image are confirmed. Compared with a non-segregated specimen, the average values of the horizontal heterogeneity coefficients of high, medium, light and fine-aggregate-segregated mixtures increased by 72.0%, 48.3%, 34.7% and 16.1%, respectively, where the change range is in accordance with the segregation degrees of several mixtures. The indirect tensile strength of fine-aggregate-, light-, medium- and high-segregated asphalt mixtures decreased by 8.3%, 16.7%, 25.0% and 45.8%, respectively, when compared with the non-segregated asphalt mixture. The index of the vertical heterogeneity coefficient has good correlation with the indirect tensile strength of segregated asphalt mixtures. The construction quality homogeneity of asphalt pavement in different regions can be reliably evaluated by the horizontal heterogeneity coefficient and vertical heterogeneity coefficient.

## 1. Introduction

An asphalt mixture is a three-dimensional multiscale material composed of aggregates, voids and asphalt mortar, and is the important building material for pavement [1]. The instability of construction quality heterogeneity is caused by the variation in the material composition, material sources and process parameters, which usually occurs in asphalt pavement. Under the influence of vehicle loads and environmental factors, the local damage to non-uniform parts appears first, and then, develops into the damage of large areas, such as looseness, hollow and rutting damage [2,3]. The microstructure formed by aggregates, voids and asphalt mortar is an important feature that determines the construction homogeneity of the asphalt mixture [4]. The difference in microstructure characteristics of various components in the asphalt mixture is the direct reflection of the construction quality homogeneity.

In recent years, the digital image processing (DIP) technology has developed quickly. Some scholars tried to assess the microstructure characteristics of asphalt mixtures according to the distribution of aggregates or voids. Zhang et al. [5] selected the industrial computer tomography (CT) device to obtain the tomography image of aggregates inside the pavement, and evaluated the distribution characteristics of aggregates via MATLAB software. Peng et al. [6] analyzed the horizontal homogeneity of asphalt mixtures at various depths in a compacted Marshall specimen and a Superpave Gyratory Compactor (SGC) specimen. The influence of aggregate homogeneity on indirect tensile strength (ITS) of asphalt mixtures was also analyzed. Based on CT scanning technology, Zhang et al. [7] put forward the indexes such as the variation range of *X*-axis inertia and aggregate centroids to evaluate the horizontal and vertical distribution homogeneity of aggregates in asphalt pavement. Bessa et al. [8] utilized the image processing technology to analyze the contact point, particle direction and segregation of aggregates in asphalt mixtures. Li et al. [9] studied the distribution homogeneity of virgin and RAP aggregates in recycling asphalt mixtures by using the DIP technique. Bruno et al. [10] used various kinds of image segmentation methods to obtain the information of aggregate gradation of asphalt mixtures and recommended the index to assess the segregation degree of aggregates. Li et al. [11] put forward a new method to study the movement of coarse aggregates and densification parameters of asphalt mixtures in the process of gyratory compaction. Cong et al. developed a real-time detection method to evaluate the segregation of compacted asphalt pavement by processing the images of paved asphalt mixtures [12].

Zhang used the image processing technology to analyze the distribution characteristics of voids in a CT image of asphalt mixtures [13]. Tan et al. [14,15] realized the digital calculation of the void ratio of asphalt mixtures based on industrial CT equipment, and analyzed the spatial distribution of voids in different asphalt mixtures. Yu et al. [16] studied the void distribution characteristics inside the Marshall specimen and SGC specimen of cold in-place recycled (CIR) asphalt mixtures. The voids were found to be unevenly distributed in asphalt mixtures. Xiao et al. [17,18] obtained the parameters for the internal structure of drainage asphalt mixtures based on industrial CT. The differential substances of asphalt mixtures were identified via the improved Nobuyuki Otsu (OTSU) method, and the spatial distribution characteristic of voids was studied. Mahmud et al. [19] studied the void properties of laboratory-fabricated porous asphalt mixtures by employing the virtual cut section (imaging technique). Li et al. [20] investigated the spatial distribution parameters of voids in mixture specimens compacted in the field and in a laboratory, and recommended the laboratory compaction method of Superpave Gyratory Compactor (SGC). Zhang et al. [21] studied the internal structure of asphalt mixtures with different gradations. The void volume inside the specimen was found to mainly be concentrated in the range from 0 mm^3^ to 50 mm^3^. Wang et al. identified the voids and obtained the accurate void size of porous asphalt mixtures based on image processing technology and the OTSU method [22].

The above studies individually investigated the aggregate or void distribution of asphalt mixtures based on DIP technology. The transition from the macrostructure to microstructure of asphalt pavement was realized, which provided some references for further understanding the structural characteristics of asphalt mixtures. However, due to the uneven distribution of brightness in CT images of pavement core samples, separating various components in the process of producing CT images effectively is difficult, and the segmentation efficiency is not high. The adhesion of aggregate particles has not been effectively solved. Meanwhile, the evaluation of the microstructure of asphalt mixtures was mostly carried out through the qualitative description of component distribution, such as aggregates and voids. The related evaluation methods usually started from a single component, and ignored the interaction between various components, which was lacking the judgment standard of heterogeneity. The variation in microstructure characteristics reflects the heterogeneity of asphalt mixtures, but there is no quantitative relationship between the two. Therefore, it is necessary to put forward a reasonable evaluation method of heterogeneity from a microstructure viewpoint to realize the quantitative evaluation of the construction homogeneity of asphalt mixtures.

Therefore, the industrial CT equipment was selected in this paper to obtain the tomography image of the core sample of asphalt pavement. According to the characteristics of CT images, an improved threshold separation algorithm based on annular partition and OTSU threshold segmentation was proposed to realize the effective segmentation of aggregate particles, voids and asphalt mortar. Based on the spatial distribution of aggregates, voids and asphalt mortar, and the area ratio of each component inside the pavement, the evaluation method for the component distribution homogeneity of asphalt pavement was proposed, and the validity of evaluation indexes was also verified. A flow chart for the methodology of this research is shown in Figure 1. The research results can provide some references for the detection and evaluation of the construction quality homogeneity of asphalt pavement.

## 2. Separation of Mixture Components for Asphalt Pavement

### 2.1. Acquisition of CT Image for Core Sample

The industrial CT equipment can virtually scan the object without damaging the materials and structure. In the form of two-dimensional and three-dimensional images, the internal structure, material composition and defects of the target object will be intuitively and accurately displayed. Then, the quantitative measurement and analysis of the internal structure of the object will be realized [5]. The industrial CT system mainly includes a ray source, detector, collimator, sample-scanning system, computer system, data-acquisition system and auxiliary system. Among these, the ray source of the industrial CT system is usually an X-ray, and the schematic diagram of a typical CT system is shown in Figure 2.

The industrial CT imaging system of the road laboratory of Chang’an University was used to obtain the tomography image of the pavement core sample. The equipment model of the industrial CT system was the Y. CT Precision S, as shown in Figure 3. The ray source of the tomography imaging system of Y. CT was a micro-focus ray source, whose resolution can achieve a um level and the spacing distance of tomography images reaches 0.1 mm. The industrial CT equipment is suitable for the high-precision detection of defects and can realize the non-destructive scanning of the structural characteristics of pavement core samples, as shown in Figure 4.

### 2.2. Separation of Various Components in CT Image

To improve the component separation effect in the CT image of the core sample of asphalt pavement, an improved threshold separation algorithm based on annular partition and OTSU threshold segmentation, according to the characteristic of uneven brightness in a CT image, was recommended [23]. The steps of the proposed algorithm are as follows:(1)Annular partition of CT image for pavement core sample.

The brightness distribution of various components in the CT image of an asphalt mixture is uneven. Before the separation of the CT image, the image needs to be divided into various regions to reduce the brightness difference at different positions in the same area. Since the CT image of a core sample is dark in the middle and the brightness around the center increases gradually, it is appropriate to divide the image into annular regions, which can reduce the difference of brightness in the same region. Here, five rings were taken as an example, and the CT image was divided into six sub-images of the central circle and five rings, as shown in Figure 5.

(2)Separation of voids in CT image.

The gray values of the aggregates and asphalt mortar in the CT image are very close. If the aggregates and asphalt mortar are regarded as a whole, there are obvious differences between the gray values of the aggregates, asphalt mortar and voids. The voids can be, firstly, separated from the aggregates and asphalt mortar in the CT image. The OTSU threshold separation algorithm was selected to process the divided sub-images, and then the segmented sub-images were combined to obtain the target separation image of the aggregates and asphalt mortar, as shown in Figure 6a. After inverting Figure 6a, the voids in the CT image can be identified, as shown in Figure 6b.

(3)Separation of aggregate in CT image.

As can be seen from Figure 6a, since the gray values of aggregates and asphalt mortar in the CT image of a core sample are close, the edge of the aggregates and asphalt mortar after the first annular separation is not segmented clearly. Therefore, the grayscale image after one threshold separation was obtained by multiplying Figure 6a with the original Figure 4, as shown in Figure 7.

Next, repeat the operation of annular partition and OTSU threshold segmentation for the grayscale image after one annular separation. The sub-images after multiple separations were assembled, as shown in Figure 8.

It can be seen from Figure 8 that after multiple annular separations, the edge of aggregate particles in the CT image has been segmented, but there are many black dots on the coarse aggregates. The black points in the image were treated by inverse area filtering; here, the inverse of Figure 8 is shown in Figure 9a. In the process of inverse area filtering, a suitable threshold was set to filter the small white area on the aggregates. The effect of filtering on aggregates is shown in Figure 9b.

Then, the separation effect of aggregate particles in the CT image of the pavement core sample would be acquired by reversing Figure 9b, as shown in Figure 10.

(4)Separation of asphalt mortar in CT image.

The asphalt mortar in the CT image can be obtained by multiplying Figure 6a with Figure 9b, as shown in Figure 11.

(5)Verification of component separation effect in CT images.

The segmented image of aggregates, voids and asphalt mortar was superimposed after being processed on the original image of Figure 4. The superimposed images are shown in Figure 12.

As shown in Figure 12, the advantage of the improved OTSU threshold separation method is more obvious than the traditional methods. The aggregates, voids and asphalt mortar in CT images can be effectively segmented by annular partition combined with the OTSU threshold separation algorithm. The boundary of different components in the CT image is clear and continuous, and there is no component information missing in the CT image of asphalt pavement. The effectiveness of the OTSU algorithm in segmenting the aggregates and voids of the asphalt mixture has also been verified in some of the literature [24,25,26]. Obviously, segmenting different components in a CT image of a pavement core sample by the improved OTSU threshold separation method is reliable.

## 3. Evaluation Index of Component Homogeneity for Asphalt Pavement

During the analysis process of CT images, it is necessary to extract and analyze the image features of target objects. The purpose of the image process in this paper is to reflect the distribution homogeneity of aggregation, voids and asphalt mortar in asphalt pavement. For the asphalt pavement that is constructed evenly, the different components should be distributed evenly in horizontal and vertical directions. Under the ideal condition, the pavement components in some local regions will not be too much or too little, and there is no agglomeration phenomenon. Generally, the geometrical features of aggregates in asphalt mixtures can be described by three independent components: form, angularity, and surface texture [27]. Among which, the form and angularity of aggregates has an important influence on the distribution homogeneity of the asphalt mixture. For an aggregate particle, its shape is the closest to an ellipse [28]. The area of the equivalent ellipse is equal to the aggregate area. According to the area and position of aggregate particles in a CT image, the homogeneity of aggregate distribution in the asphalt mixture can be reflected [7]. Therefore, the area parameters of aggregates, voids and asphalt mortar in a CT image of a core sample were selected to analyze the distribution homogeneity of various components in asphalt pavement.

### 3.1. Evaluation Index of Component Homogeneity

To reasonably evaluate the distribution homogeneity of aggregates, voids and asphalt mortar in the CT image of the pavement core sample, the horizontal tomography image scanned by the industrial CT equipment was symmetrically divided into four regions, as shown in Figure 13.

In this paper, the component area ratio (AR) of asphalt pavement was defined as the ratio of the area of aggregates, voids, asphalt mortar and the whole image area. The self-created functions in MATLAB R2019b were used to calculate the area ratios of aggregates, voids and asphalt mortar in four regions. The horizontal heterogeneity coefficient (*U_H_*) for various components was recommended to evaluate the distribution homogeneity of aggregates, voids and asphalt mortar of asphalt pavement in the horizontal direction in the tomography image. The definition of the *U_H_* index is the variation degree of the area ratio of different components in four regions in the same tomography image, as shown in Equation (1).
(1)$$U_{H}=\sum_{i=1}^{m}U_{Hi}\times f_{i}$$

Here, *U_Hi_* is the horizontal distribution homogeneity index of the *i*-th component in the CT image; *i* = 1, 2, 3, where 1 represents aggregates, 2 represents voids and 3 represents asphalt mortar, of which the formula is shown in Equation (2).

*f_i_* is the weight coefficient of the horizontal distribution homogeneity index for the *i*-th component in the CT image, of which the formula is shown in Equation (3).
(2)$$U_{Hi}=\sqrt{\frac{\sum_{j=1}^{n}{\left(AR_{ij}-\bar{AR_{i}}\right)}^{2}}{n-1}}/\frac{\sum_{j=1}^{n}AR_{ij}}{n}$$

Here, *AR_ij_* is the area ratio of the *i*-th component in the *j*-th region in the tomography image.

${\bar{AR}}_{i}$ is the average value of the area ratio of the *i*-th component in different regions in the tomography image.

*n* is the number of regions in the tomography image; here, it is 4.

Generally, the smaller the horizontal heterogeneity coefficient of mixture components, the more homogeneous the asphalt pavement is in the horizontal direction. The horizontal heterogeneity coefficient of *U_H_* reflects the distribution homogeneity of constituent components in one tomography image of the asphalt pavement; then, the weak areas of construction homogeneity at different depths of the pavement can be determined.

Meanwhile, to quantitatively express the vertical distribution homogeneity of constituent components in different tomography images of asphalt pavement, the vertical heterogeneity coefficient of *U_V_* is defined as the area ratio variation degree of the core sample in the vertical direction in different tomography images. The calculation formula of *U_V_* is shown in Equation (3).
(3)$$U_{V}=\sum_{i=1}^{m}U_{Vi}\times f_{i}^{\prime }$$

Here, *U_Vi_* is the vertical distribution homogeneity index of the *i*-th component; *i* = 1, 2, 3, where 1 represents aggregates, 2 represents voids and 3 represents asphalt mortar, of which the formula is shown in Equation (4).

*f_i_*^′^ is the weight coefficient of the vertical distribution homogeneity index for the i-th component of the pavement core sample, of which the formula is shown in Equation (5).
(4)$$U_{Vi}=\sqrt{\frac{\sum_{k=1}^{p}{\left(AR_{ik}-{{\bar{AR}}_{i}}^{\prime }\right)}^{2}}{p-1}}/\frac{\sum_{k=1}^{p}AR_{ik}}{p}$$
(5)$$f_{i}^{\prime }=\frac{\sum_{k=1}^{p}AR_{ik}}{\sum_{i=1}^{3}\sum_{k=1}^{p}AR_{ik}}$$

Here, *AR_ik_* is the area ratio of the *i*-th component in the *k*-th tomography image.

${{\bar{AR}}_{i}}^{\prime }$ is the average value of the area ratio of the *i*-th component in different tomography images of the core sample.

*P* is the number of tomography images of the core sample. For a core sample with a height of 60 mm, a total of 6 tomography images are selected at 10 mm intervals within the height range from 5 mm to 55 mm.

The vertical heterogeneity coefficient of *U_V_* reflects the vertical distribution homogeneity of aggregates, voids and asphalt mortar in asphalt pavement. The smaller the *U_V_* value, the more uniform the constituent component distribution in the vertical direction is. Namely, the construction quality homogeneity of the asphalt pavement is better.

The horizontal and vertical distribution uniformity of mixture components are two important aspects that reflect the construction uniformity of asphalt pavement. Based on the horizontal heterogeneity coefficient and vertical heterogeneity coefficient of mixture components, the construction heterogeneity coefficient of *U* was proposed to comprehensively evaluate the construction quality of asphalt pavement, as shown in Equation (6).
(6)$$U=c_{1}U_{H}+c_{2}U_{V}$$

Here, *c*_1_ and *c*_2_ are both 0.5.

### 3.2. Verification of the Validity of Evaluation Index

#### 3.2.1. Specimen Preparation of Segregated Asphalt Mixture

Referring to some of the relevant literature [5,29], and according to the characteristics of segregated asphalt pavement, the indexes of the passing rate of the key sieve, asphalt content and void ratio were selected to classify the mixtures into five segregation levels such as non-segregation, light segregation, medium segregation, high segregation and fine-aggregate segregation. Among these, the light level (L), medium level (M) and high level (H) of segregation belong to coarse-aggregate segregation.

The AC-20 asphalt mixture with non-segregation was divided into three parts of A, B and C by the passing ratio of the key sieve of 4.75 mm and 9.5 mm. Among these, A is the part of the asphalt mixture with an aggregate size smaller than 4.75 mm. B is the part of the asphalt mixture which has an aggregate size in the range from 4.75 mm to 9.5 mm. C is the part of the asphalt mixture which has an aggregate size larger than 9.5 mm.

The mixture with various segregation levels was prepared by adjusting the proportion of parts A, B and C. The asphalt content in the mixture was measured by using the combustion method in the laboratory. The asphalt mixture after being burnt was also screened to determine the aggregate gradation. The passing ratio, void ratio and asphalt content of asphalt mixtures with various segregation levels are shown in Table 1 and Table 2.

It was verified that the variation range of the aggregate gradation, asphalt content and void ratio of asphalt mixtures with five segregation degrees could satisfy the classification requirement [5].

Then, the Marshall specimens of five kinds of segregated asphalt mixtures, such as light, medium, high, fine aggregation and non-segregation were formed. The specimens were scanned from the top to the bottom by using industrial CT equipment. The CT images were selected to analyze the distribution homogeneity of the components of aggregates, voids and asphalt mortar in the asphalt mixture with various segregation levels.

#### 3.2.2. Validity of Evaluation Index

To reduce the influence of the poor imaging effect at both ends of the Marshall specimen, the tomography images were selected at 10 mm intervals within the height range from 5 mm to 55 mm as the study objects. The CT images were processed by using MATLAB software to calculate the indexes of horizontal and vertical heterogeneity coefficients of constituent components in the asphalt mixtures.

Firstly, the distribution of constituent components in the tomography image of the asphalt mixture was observed. Whether the horizontal distribution homogeneity coefficient of constituent components is consistent with the observation results was analyzed to verify the reliability of the horizontal heterogeneity coefficient index. For the Marshall specimens of the asphalt mixture with five segregation degrees, four CT images of every specimen were selected along the height direction (15 mm/25 mm/35 mm/45 mm), as shown in Figure 14. The calculated results of the horizontal heterogeneity coefficient of constituent components of asphalt mixtures in different CT images are shown in Table 3.

As shown in Figure 14, there are some differences in the component distribution of aggregates, voids and asphalt mortar in various CT images of the same Marshall specimen. For the fine-aggregate-segregated asphalt mixture, the order of component homogeneity in various tomography images is 3 > 2 > 1 > 4. For the non-segregated asphalt mixture, the order of component homogeneity in different tomography images is 2 > 3 > 4 > 1. For the light-segregated asphalt mixture, the order of component homogeneity in different tomography images is 2 > 3 > 1 > 4. For the medium-segregated asphalt mixture, the order of component homogeneity in different tomography images is 2 > 3 > 4 > 1. For the high-segregated asphalt mixture, the order of component homogeneity in different tomography images is 2 > 3 > 1 > 4. In the four tomography images which were selected, the order of component distribution homogeneity at different heights of several segregated mixture specimens is not consistent.

In Table 3, the order of horizontal heterogeneity coefficients of various components in different tomography images of five kinds of segregated asphalt mixtures calculated by MATLAB image processing technology is consistent with the observation results in Figure 14. In addition, the rank of average values of horizontal heterogeneity coefficients of the components in four tomography images of several segregated asphalt mixtures is H > M > L> F > N. Compared with the non-segregated specimen, the average values of horizontal heterogeneity coefficients of high-, medium-, light- and fine-aggregate-segregated specimens increased by 72.0%, 48.3%, 34.7% and 16.1%, respectively. The change range of the average value of the horizontal heterogeneity coefficient is in accordance with the segregation degrees of several mixtures. Thus, the index of the horizontal heterogeneity coefficient of *U_H_* has good reliability to evaluate the component homogeneity at different depths of asphalt pavement.

Then, the image processing and statistical analysis of various tomography images for the same specimen were conducted to calculate the index of vertical heterogeneity coefficients of asphalt mixtures with different segregation degrees. Meanwhile, the mixture specimens after CT scanning were used to carry out the indirect tensile test at the temperature of 25 °C and the loading rate of 50 mm/min. The regression analysis of the vertical heterogeneity coefficients of components and indirect tensile strength of asphalt mixtures with different segregation degrees was conducted by using SPSS software (version 26.0), as shown in Figure 15.

It can be seen from Figure 15 that there are significant differences in vertical heterogeneity coefficients of asphalt mixtures with various segregation degrees. Among the several asphalt mixtures, the heterogeneity coefficient of the vertical distribution of the constituent component of the non-segregated asphalt mixture is the smallest. Compared with the non-segregated asphalt mixture, the vertical heterogeneity coefficient of the constituent component of the fine-aggregate-, light-, medium- and high-segregated asphalt mixtures increased by 6.5%, 11.7%, 18.2% and 55.8%, respectively. Obviously, with the increase in the segregation degree of the coarse aggregate of the asphalt mixture, the vertical heterogeneity coefficient of the constituent component gradually increases. The vertical distribution of the constituent component of the fine-aggregate-segregated asphalt mixture is superior to the light-segregated asphalt mixture.

In addition, the indirect tensile strength of the asphalt mixture decreases by various extents when the aggregate segregation occurs. The influence trend of aggregate segregation on the mechanical properties of the asphalt mixture is consistent with the previous studies [30,31,32]. Among which, the indirect tensile strength of the fine-aggregate-segregated asphalt mixture is higher than that with light segregation. Compared with the non-segregated asphalt mixture, the indirect tensile strength of fine-aggregate-, light-, medium- and high-segregated asphalt mixtures decreased by 8.3%, 16.7%, 25.0% and 45.8%, respectively, which is consistent with the change rule of the vertical heterogeneity coefficient of component distribution. Namely, with the increase in the segregation degree of the coarse aggregate, the indirect tensile strength of the segregated asphalt mixture decreases gradually.

The fitting formula, consisting of the indirect tensile strength and vertical heterogeneity coefficient of asphalt mixtures with different segregation degrees, is y = 295.22 × ^2^ − 71.14x + 4.93, which has good correlation. Therefore, it is reasonable to evaluate the distribution homogeneity of constituent components of different asphalt mixtures by using the index of the vertical heterogeneity coefficient.

The above analysis shows that the index of the horizontal heterogeneity coefficient of the constituent component could quantitatively evaluate the construction quality homogeneity of asphalt pavement at different depths. Based on the index of the vertical heterogeneity coefficient, the variation in the construction quality homogeneity in different regions of asphalt pavement can be reflected. Some studies also confirmed that the quantity distribution and location distribution of aggregate particles could effectively characterize the homogeneity of asphalt mixtures [33,34].

## 4. Detection and Evaluation of Component Homogeneity for Asphalt Pavement

Relying on the construction site of a medium layer of Xia-Guang Avenue in Fuyang, Anhui province, two sections with the length of 1000 m were selected as the detection section, which are called section Ⅰ and section Ⅱ. In section Ⅰ and Ⅱ, the density of various regions was measured by using a non-nuclear densitometer of PQI with the transverse space of 1.5 m and the longitudinal space of 50 m. The maximum theoretical density of asphalt mixtures of section Ⅰ and section Ⅱ was determined by checking the experimental data from the site laboratory. According to the density data of various areas of section Ⅰ and section Ⅱ detected by PQI, the percentage of high-, medium- and low-density areas of detection points in the two sections was analyzed, as shown in Table 4.

In section Ⅰ and Ⅱ, 10 points were selected as the representative points to analyze the density distribution of the asphalt pavement. According to the ratio of points of a high, medium and low density in Table 4, one point in the high-density region, seven points in the medium-density region and two points in the low-density region were required for section Ⅰ and section Ⅱ, which ensures that the selected points can represent the density distribution homogeneity of the whole detection section.

The core samples with the height of 60 mm were drilled at the representative points in section Ⅰ and section Ⅱ, and the core samples were scanned by using the industrial CT system. Six tomography images were selected at 10 mm intervals in the height range from 5 mm to 55 mm. Image processing and analysis of the selected images were conducted. The construction quality homogeneity of asphalt pavement in the corresponding area was evaluated by comparing the average values of the component heterogeneity coefficients of six selected images from the core sample. The component heterogeneity coefficients of typical core samples in section Ⅰ and section Ⅱ are shown in Table 5.

It can be seen from Table 5 that there is no obvious boundary between the horizontal heterogeneity coefficient, vertical heterogeneity coefficient and construction heterogeneity coefficient of the core samples selected in the high-, medium- and low-density areas in section Ⅰ and section Ⅱ. Among the different core samples in the three kinds of selected regions, the correlation between the indexes of the heterogeneity coefficient and the density of the core sample is not regular. It is unreliable that the greater the density of the core sample, the better or worse the homogeneity of the component distribution inside the asphalt pavement is. There is no direct relationship between the pavement density and segregation degree of the constituent component of asphalt pavement. Therefore, it is not sufficient to evaluate the compaction homogeneity of asphalt pavement by only depending on the degree of compaction. According to the difference in pavement component distribution, it is reasonable to evaluate the construction quality homogeneity of asphalt pavement from the viewpoint of the microstructure by the indexes of the horizontal heterogeneity coefficient, vertical heterogeneity coefficient and construction heterogeneity coefficient.

According to the data in Table 5, the statistics of the horizontal heterogeneity coefficient, vertical heterogeneity coefficient and construction heterogeneity coefficient of typical core samples in section Ⅰ and section Ⅱ were calculated, as shown in Table 6.

As shown in Table 6, the horizontal heterogeneity coefficient, vertical heterogeneity coefficient and construction heterogeneity coefficient of components between different representative core samples in section Ⅰ and Ⅱ have certain variability. Compared with section Ⅱ, the average value and variation coefficient of the heterogeneity coefficient of representative core samples in section Ⅰ are smaller, and thus, the component distribution homogeneity of section Ⅰ is better than section Ⅱ. Therefore, the construction quality homogeneity of section Ⅰ is better than section Ⅱ from the component distribution of asphalt pavement.

## 5. Conclusions

According to the characteristic of uneven brightness of the CT image of a core sample from asphalt pavement, an improved threshold separation algorithm based on annular partition and OTSU threshold segmentation is proposed. The aggregates, voids and asphalt mortar in CT images can be effectively segmented by annular partition combined with the OTSU threshold separation algorithm. By superimposing the segmented image of aggregates, voids and asphalt mortar onto the original image, the segmentation and identification effect of aggregates, voids and asphalt mortar in the CT image is confirmed.

Based on the distribution homogeneity of aggregates, voids and asphalt mortar in the horizontal direction in the CT image, the horizontal heterogeneity coefficient is proposed to reflect the construction quality homogeneity of the core sample at different depths in asphalt pavement; thus, the weak areas of the construction quality homogeneity of the core sample can be determined. Compared with the non-segregated specimen, the average values of horizontal heterogeneity coefficients of high-, medium-, light- and fine-aggregate-segregated mixtures increased by 72.0%, 48.3%, 34.7% and 16.1%, respectively, in which the change range is in accordance with the segregation degrees of several mixtures. The index of the horizontal heterogeneity coefficient has good reliability to evaluate the component homogeneity at different depths of asphalt pavement.

Based on the distribution homogeneity of aggregates, voids and asphalt mortar in the vertical direction in the CT image, the vertical heterogeneity coefficient is proposed to reflect component distribution homogeneity of the core sample. The indirect tensile strength of fine-aggregate-, light-, medium- and high-segregated asphalt mixtures decreased by 8.3%, 16.7%, 25.0% and 45.8%, respectively, when compared with the non-segregated asphalt mixture. The index of the vertical heterogeneity coefficient has good correlation with the indirect tensile strength of segregated asphalt mixtures. The construction quality homogeneity of asphalt pavement in different regions can be reliably evaluated by the vertical heterogeneity coefficient.

## Figures and Tables

**Figure 1 materials-15-07284-f001:**
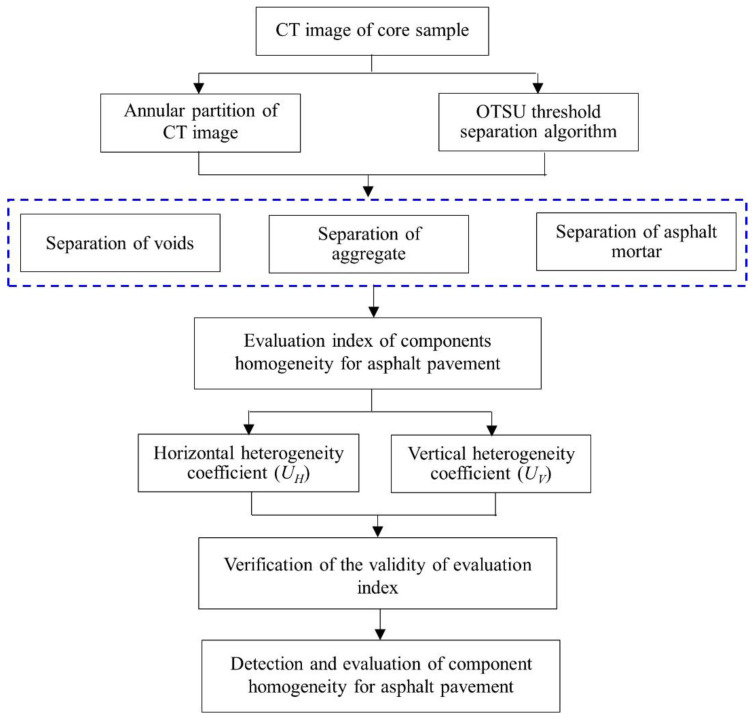
The flow chart for the research methodology.

**Figure 2 materials-15-07284-f002:**
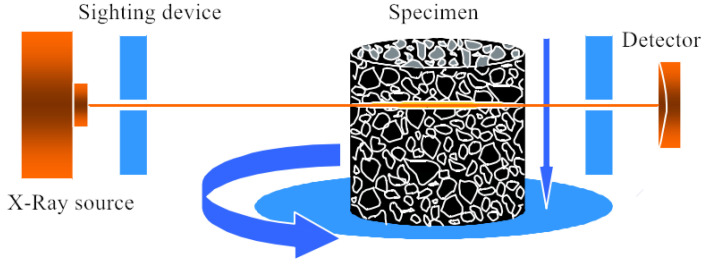
Principle of typical CT system.

**Figure 3 materials-15-07284-f003:**
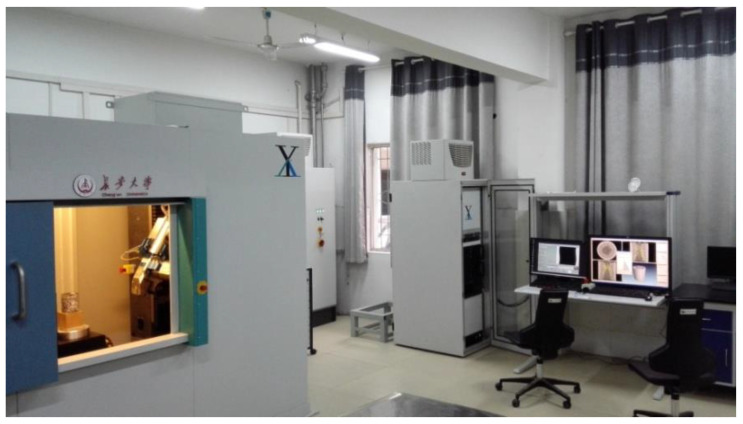
Industrial computed tomography imaging system.

**Figure 4 materials-15-07284-f004:**
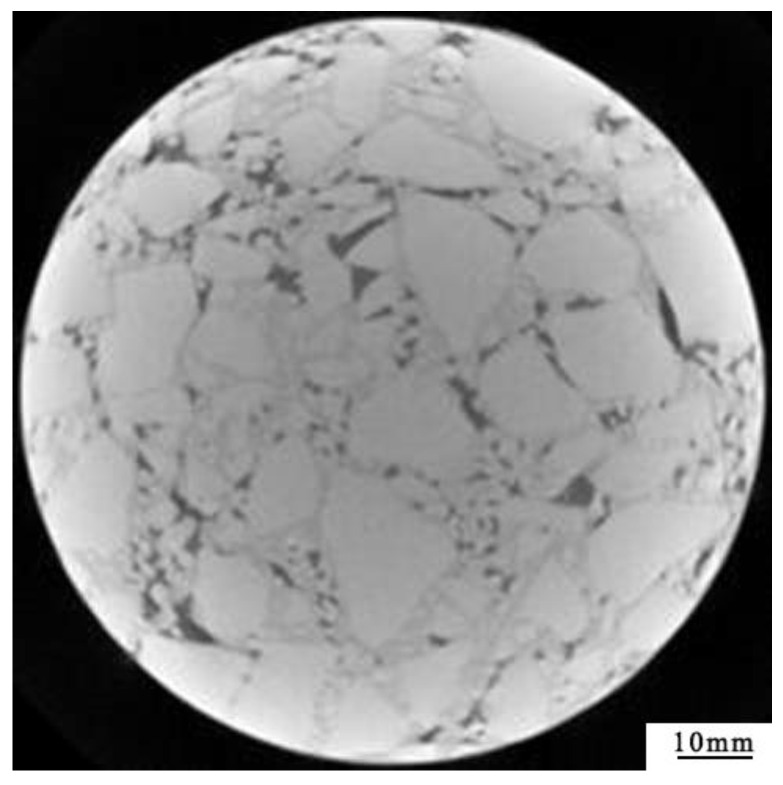
CT image of core sample for asphalt pavement.

**Figure 5 materials-15-07284-f005:**
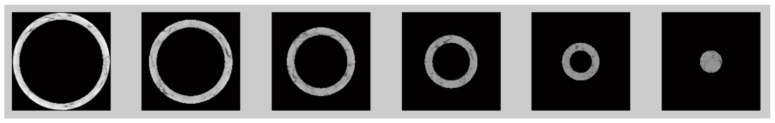
Six sub-images of CT image after annular partition.

**Figure 6 materials-15-07284-f006:**
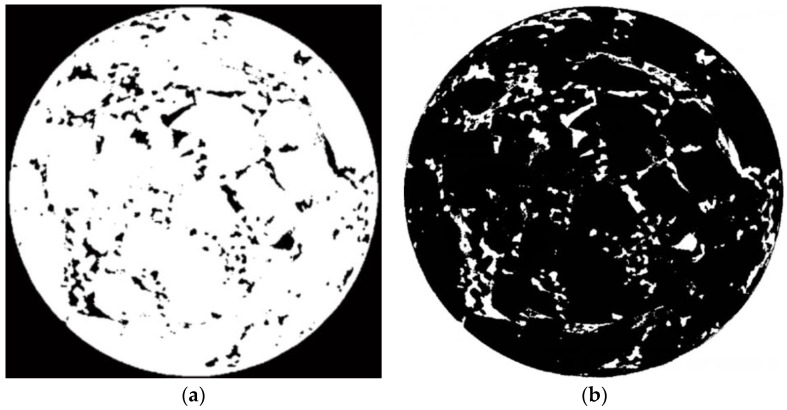
Void separation of CT image. (**a**) Aggregates and asphalt mortar. (**b**) Voids.

**Figure 7 materials-15-07284-f007:**
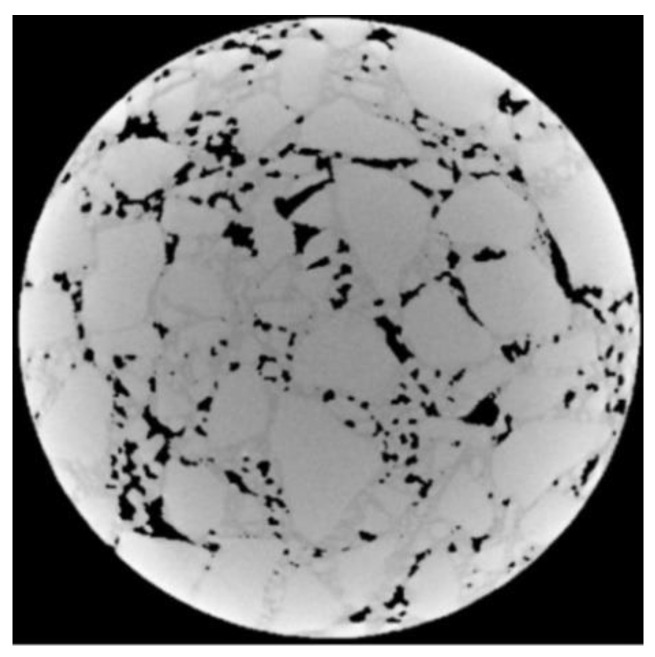
Grayscale image after one annular separation.

**Figure 8 materials-15-07284-f008:**
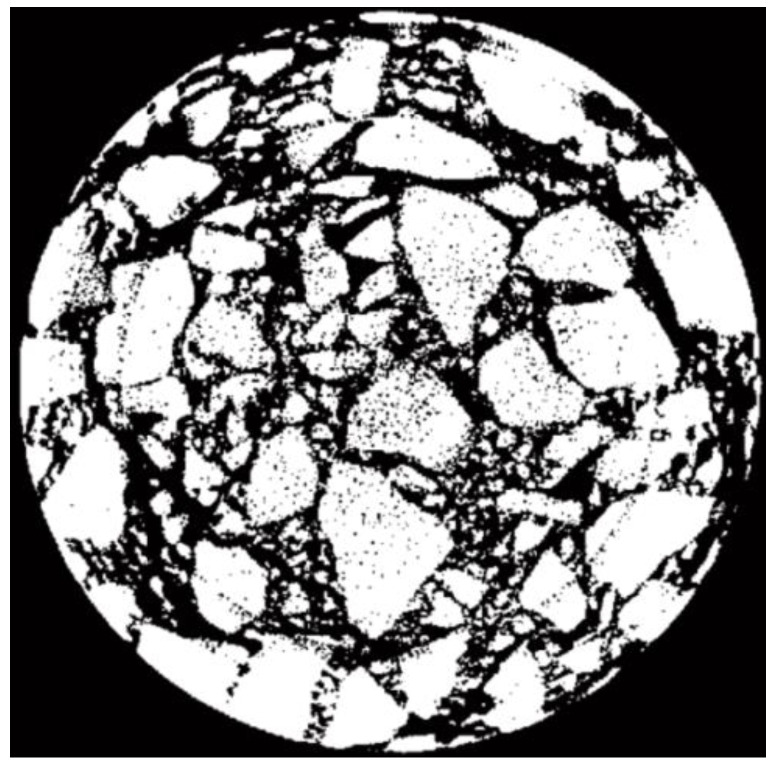
The effect after multiple annular separations.

**Figure 9 materials-15-07284-f009:**
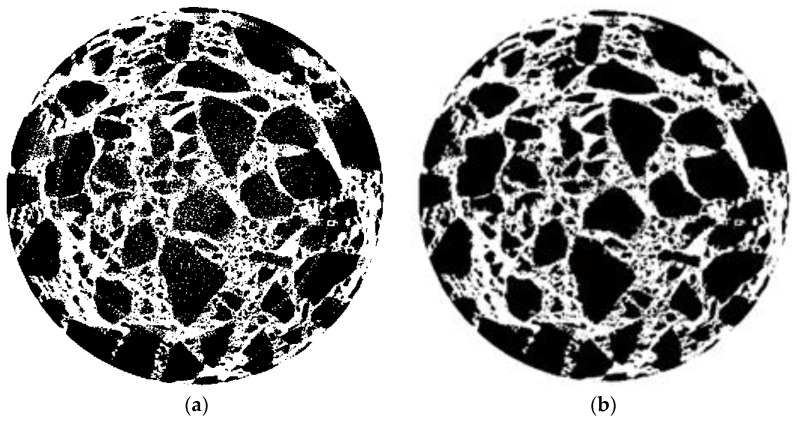
Image after being processed by inverse area filtering. (**a**) Inverted image. (**b**) Filtered image.

**Figure 10 materials-15-07284-f010:**
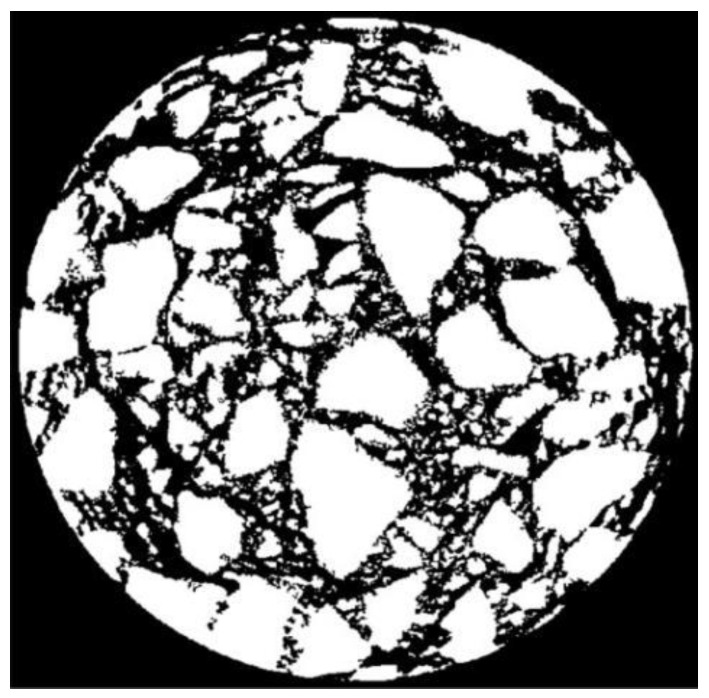
Separation effect of aggregates in CT image.

**Figure 11 materials-15-07284-f011:**
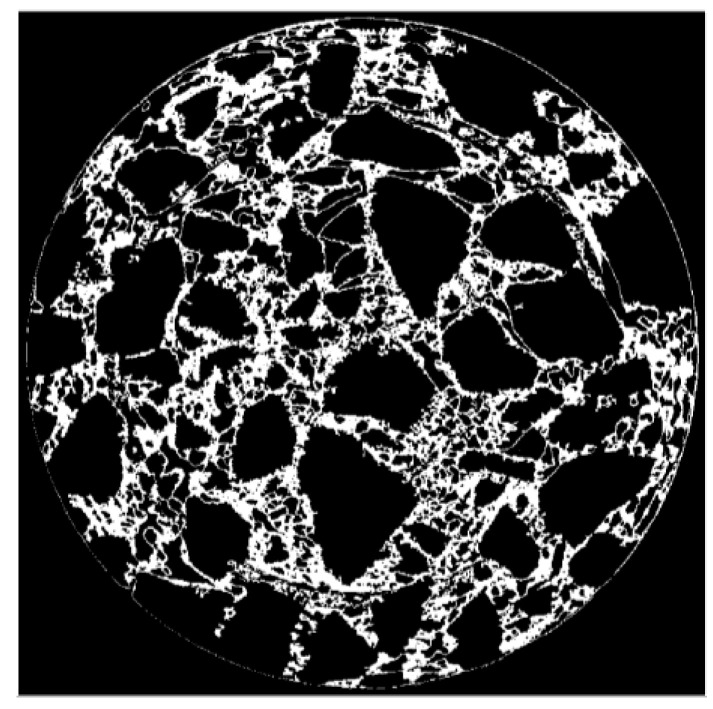
Separation effect of asphalt mortar in CT image.

**Figure 12 materials-15-07284-f012:**
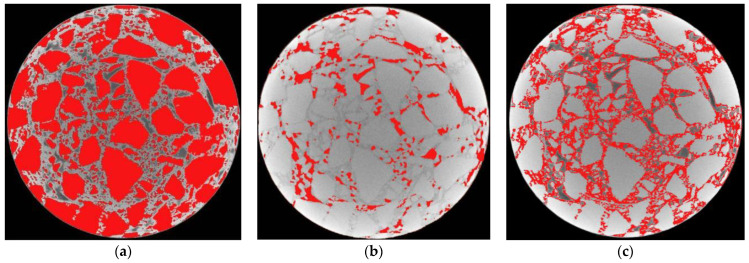
Validation of the separation effect of various components. (**a**) Aggregates. (**b**) Voids. (**c**) Asphalt mortar.

**Figure 13 materials-15-07284-f013:**
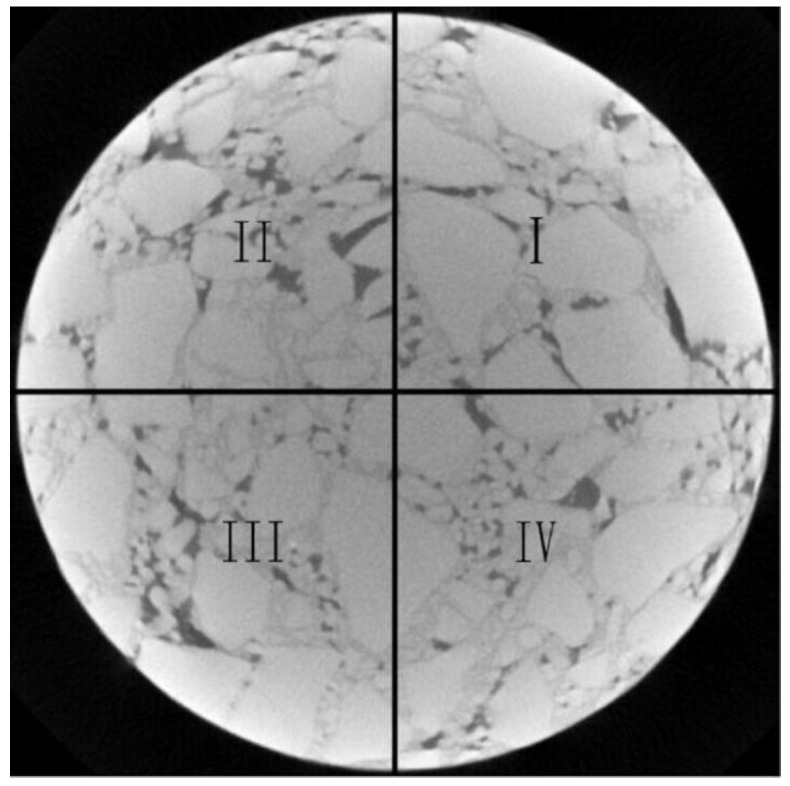
The region division of CT image. (I–IV respectively refers to the first to fourth regions of CT image.)

**Figure 14 materials-15-07284-f014:**
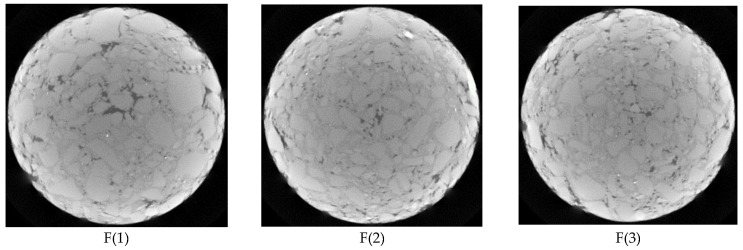
CT images of asphalt mixtures with different segregation degrees.

**Figure 15 materials-15-07284-f015:**
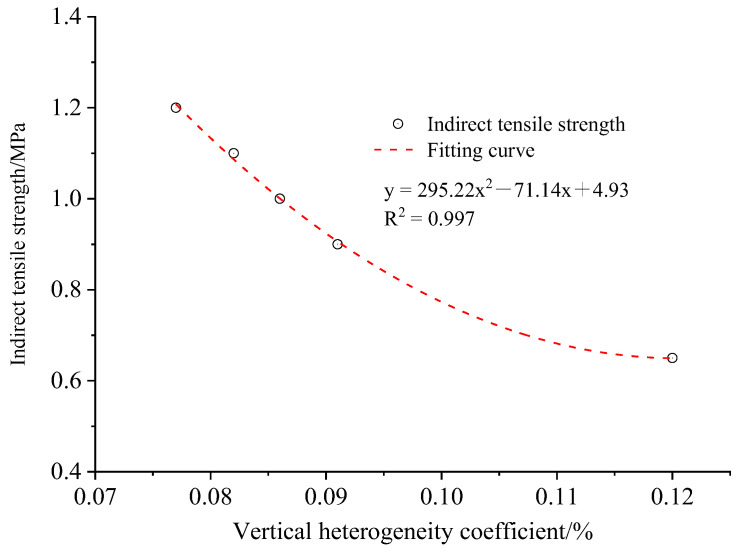
Relationship between vertical heterogeneity coefficient and indirect tensile strength.

**Table 1 materials-15-07284-t001:** Gradation composition of AC-20 segregated mixture.

Sieve Size/mm	Passing Ratio/%				
	N	L	M	H	F
26.5	100	100.0	100.0	100.0	100.0
19	98.2	99.0	98.8	98.5	99.3
16	89.2	87.2	84.6	80.9	90.9
13.2	76	71.2	65.2	56.9	79.5
9.5	62.2	56.3	47.4	35.0	68.7
4.75	37	32.8	27.7	21.5	41.3
2.36	26.7	24.0	21.3	17.9	28.1
1.18	18.3	18.2	17.0	15.4	19.8
0.6	12.6	14.6	13.9	12.9	15.5
0.3	8.1	9.8	9.7	9.5	10.0
0.15	5.9	6.2	6.0	5.7	6.5
0.075	5	4.3	4.1	3.8	4.6

**Table 2 materials-15-07284-t002:** Void ratio and asphalt content of AC-20 segregated mixtures.

Mixture Type	Void Ratio/%	Asphalt Content/%
N	6.5	3.90
L	8.9	3.52
M	11.4	3.13
H	13.5	2.88
F	3.1	4.25

**Table 3 materials-15-07284-t003:** Horizontal heterogeneity coefficients of different CT images.

Section Number	Segregation Degree				
	F	N	L	M	H
1	0.142	0.136	0.165	0.188	0.213
2	0.135	0.089	0.146	0.164	0.185
3	0.126	0.113	0.157	0.172	0.192
4	0.145	0.132	0.168	0.176	0.221
Average value	0.137	0.118	0.159	0.175	0.203

**Table 4 materials-15-07284-t004:** Proportion of different density areas on detection section %.

Detection Section	Region Type		
	High	Medium	Low
Ⅰ	9.5	68.8	21.7
Ⅱ	10.1	71.5	18.4

**Table 5 materials-15-07284-t005:** Component homogeneity index of representative core samples in detection section.

Index		Region Type									
		High	Medium							Low	
*U_H_*	Ⅰ	0.212	0.124	0.153	0.116	0.128	0.135	0.143	0.151	0.131	0.136
	Ⅱ	0.241	0.142	0.135	0.138	0.144	0.162	0.159	0.164	0.139	0.128
*U_V_*	Ⅰ	0.122	0.072	0.081	0.086	0.092	0.095	0.106	0.121	0.098	0.105
	Ⅱ	0.119	0.126	0.091	0.085	0.083	0.131	0.072	0.097	0.089	0.114
*U*	Ⅰ	0.167	0.098	0.117	0.101	0.110	0.115	0.125	0.136	0.115	0.121
	Ⅱ	0.180	0.134	0.113	0.112	0.114	0.147	0.116	0.131	0.114	0.121

**Table 6 materials-15-07284-t006:** Statistics of component distribution homogeneity of detection section.

Index	Section	Average Value	Standard Deviation	Variation Coefficient
*U_H_*	Ⅰ	0.143	0.027	0.188
	Ⅱ	0.155	0.032	0.209
*U_V_*	Ⅰ	0.098	0.016	0.166
	Ⅱ	0.101	0.020	0.201
*U*	Ⅰ	0.120	0.020	0.164
	Ⅱ	0.128	0.022	0.169

## Data Availability

Not applicable.

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
