# Peer review of "A Novel Evaluation Method of Construction Homogeneity for Asphalt Pavement Based on the Characteristic of Component Distribution"

_materials, 2022, doi:10.3390/ma15207284_

Round 1

Reviewer 1 Report

This paper studied the construction homogeneity of asphalt pavement via X-ray CT technology. Overall, it is a quite an interesting topic. As a result, there are some comments for authors that need to be considered:

1.      Overall, the quality of the writing required to be improved, I suggest that the authors should go through the manuscript carefully in case some basic mistakes.

2.      The methodology of this research should be added with a flow chart.

3.      Be careful about the abbreviation used in paper, they should be introduced when the first time being used.

4.      Please mark the scale of the material structure in figure 3.

5.      There are so many edit errors, I strongly suggested that the authors go through this manuscript carefully. For instance, equation 3 is missed, or equation 4 should be 3. The size of figure 13 is out beyond the content layout. Table 3 should be redrawn again.

6.      The results and conclusion section should go beyond summarizing the test outcome and explain the reason behind the results, and the understanding gained from those results.

Reviewer 2 Report

The work done is good and the methods applied are also satisfactory. The article is interesting and written in a reasonable manner. Some minor suggestions are:

1. Correct author names superscript

2. Write full form once "Computed tomography" scan in the main text once before abbreviation(CT).

3. Draw a table of abbreviations, symbols, etc.

4. Follow the MDPI format for mentioning figures in the text.

5. Table-3 top row needs updation.

I wish good luck to all the authors for their excellent work. 

Reviewer 3 Report

-Abstract. Please define CT and OTSU

-Abstract. Can the authors develop the most promising results (in numbers)??

-Is there any way to assess the porosity of the mixtures with the methodology proposed in the article?

-Table 3. take care in titles.

-The conclusions need greater numerical discussions than the results

-analyzes of the results need to be compared with previous findings from the literature

-To be a work with good quality of innovation, the number of references used to develop the paper are very few. Could you develop your proposal in a greater number of references?

-The reviewer believes that if algorithms were used in Matlab or another program, how could another author access it?

-Be careful with the large number of spaces that appear on some pages

Reviewer 4 Report

Dear Author(s),

The manuscript ID: Materials 2021, 14 and titled “A novel evaluation method of construction homogeneity for as-phalt pavement based on the characteristic of component distribution” was reviewed. It is a regular paper with interesting results with some experiments supported by image techniques.

The research aim is to evaluate the construction homogeneity of asphalt pavement material and apply the image technique in order to identify the effects of heterogeneity on the pavement via image technique.

Paper is comparatevily well written and it was properly explained for readers. Findings from the research were acceptably discussed considering the literature, and conclusion was consistent with generated data. 

It was evaluated that it is a regular paper and finding could attract readers especially practisinary, research engineers and modeling and analysis persons in the state of the art. The manuscript can be accepted after some revisions.

1. Please provide full name of CT image, OTSU in abstract and text. 

2. Please provide off-axis heterogeneity co-efficient considering the indexes of horizontal heterogeneity co-efficient and vertical heterogeneity coefficient of mixture components, if possible.

3. Please also do discussion on material density considering heterogeneity coefficient, if possible.  

4. Please do comparision aggregate geometry and heterogeneity coefficient in related part of the manuscript

My best regards,
